# Fine-Structure Analysis of Perhydropolysilazane-Derived Nano Layers in Deep-Buried Condition Using Polarized Neutron Reflectometry

**DOI:** 10.3390/polym12102180

**Published:** 2020-09-24

**Authors:** Kazuhiro Akutsu-Suyama, Hiroshi Kira, Noboru Miyata, Takayasu Hanashima, Tsukasa Miyazaki, Satoshi Kasai, Dai Yamazaki, Kazuhiko Soyama, Hiroyuki Aoki

**Affiliations:** 1Neutron Science and Technology Center, Comprehensive Research Organization for Science and Society (CROSS), 162-1 Shirakata, Tokai, Ibaraki 319-1106, Japan; h_kira@cross.or.jp (H.K.); n_miyata@cross.or.jp (N.M.); t_hanashima@cross.or.jp (T.H.); t_miyazaki@cross.or.jp (T.M.); s_kasai@cross.or.jp (S.K.); 2Materials and Life Science Division, J-PARC Center, Japan Atomic Energy Agency, 2-4 Shirakata, Tokai, Ibaraki 319-1195, Japan; dai.yamazaki@j-parc.jp (D.Y.); soyama.kazuhiko@jaea.go.jp (K.S.); hiroyuki.aoki@j-parc.jp (H.A.); 3Institute of Materials Structure Science, High Energy Accelerator Research Organization, 203-1 Shirakata, Tokai, Ibaraki 319-1106, Japan

**Keywords:** perhydropolysilazane, neutron reflectivity, polarization analysis

## Abstract

A large background scattering originating from the sample matrix is a major obstacle for fine-structure analysis of a nanometric layer buried in a bulk material. As polarization analysis can decrease undesired scattering in a neutron reflectivity (NR) profile, we performed NR experiments with polarization analysis on a polypropylene (PP)/perhydropolysilazane-derived SiO_2_ (PDS)/Si substrate sample, having a deep-buried layer of SiO_2_ to elucidate the fine structure of the nano-PDS layer. This method offers unique possibilities for increasing the amplitude of the Kiessig fringes in the higher scattering vector (*Q*_z_) region of the NR profiles in the sample by decreasing the undesired background scattering. Fitting and Fourier transform analysis results of the NR data indicated that the synthesized PDS layer remained between the PP plate and Si substrate with a thickness of approximately 109 Å. Furthermore, the scattering length density of the PDS layer, obtained from the background subtracted data appeared to be more accurate than that obtained from the raw data. Although the density of the PDS layer was lower than that of natural SiO_2_, the PDS thin layer had adequate mechanical strength to maintain a uniform PDS layer in the depth-direction under the deep-buried condition.

## 1. Introduction

Silica-based ceramics, which are synthesized from preceramic polymers, have been widely used as surface-coating materials to protect various components from dust, dirt, and moisture [1,2]. Perhydropolysilazane (PHPS) is a promising preceramic polymer for protecting organic and inorganic materials in corrosive and oxidative environments [3,4] because it can be easily used to synthesize high-quality silica layers on metallic materials through room-temperature hydrolysis or oxidation [5,6]. As silica-based ceramics are non-conductive and have high thermal resistance, their application as a simple separating layer, for example, as an insulating layer on semiconductor substrates, has been investigated. In fact, conductive composite networks can be successfully cut off by a separation layer made of ceramic glass inside the composite [7]. Recently, novel functional organic polymer-silica hybrid thin layer films have been developed and studied in view of their multi-functionality and applicability for a variety of processes [8,9]. Based on these trends, silica-based ceramic layers are often coated under deep-buried conditions on support materials, such as silicon or glass substrates. However, a nondestructive structural characterization of buried thin-layers is still limited and remains a challenging task.

X-ray (XR) and neutron reflectivity (NR) techniques are promising nondestructive methods to evaluate the structure of buried thin-layer samples [10,11]. However, in the XR/NR measurement of a buried thin layer sample, undesired scattering can increase, thus making it difficult to carry out a fine structural analysis of the sample because the X-ray/neutron beam passes through the sample matrix. The use of deuterium (^2^H) labeling techniques [12,13] is beneficial for controlling the neutron scattering contrast of the organic samples and reducing the proton incoherent scattering because there are significant differences in the incoherent scattering cross-sections of the two isotopes (*σ*_incoh_ (^1^H) = 79.7 × 10^−24^ cm^2^ and *σ*_incoh_ (^2^H) = 2.0 × 10^−24^ cm^2^) [14]. However, it is not suitable for organic polymer samples because it is difficult to obtain a large amount of deuterated polymer materials. Polarized neutron scattering techniques are also important and powerful tools in the study of buried thin layers and organic thin layer samples because these techniques can decrease the spin-incoherent hydrogen scattering background [15,16,17]. These techniques have been used in small-angle neutron scattering experiments; however, few studies have been reported on utilizing the polarization analysis on neutron reflectometry for background subtraction.

In this study, a fine-structure analysis of PHPS-derived silica (PDS) thin layers was conducted using NR techniques with polarization analysis. Firstly, NR measurements of basic PDS thin layers with different thicknesses were performed at the air–solid interface to investigate the structure of simple PDS thin layers. Next, polarized neutron reflectivity (PNR) measurements of an extremely thin PDS layer (approximately 10 nm), sandwiched between an Si substrate and a polypropylene (PP) plate was performed to elucidate the structural change of the PDS thin-layer under deep-buried conditions. The PP/PDS/Si substrate was a simulated sample of an insulating layer typically formed on the Si substrates. Finally, a Fourier transform analysis of the PNR data was performed to objectively evaluate the PDS layer thickness of the sample, and the peak position in the Fourier space was in good agreement with the layer thickness.

## 2. Materials and Methods

### 2.1. Materials

Silicon wafers (diameter and thickness of 5.08 cm and 0.5 mm, respectively) were supplied by Crystal Base Co. Ltd. (Osaka, Japan). A PHPS polymer (AQUAMICA, Product Code: NP110-10, amine catalyst type) and a polypropylene plate (PP plate, 50 × 50 mm^2^ and 3 mm thickness) were supplied by AZ Electronic Materials Co., Ltd. (Tokyo, Japan) and As-One Co., Ltd., (Tokyo, Japan), respectively. The PP plate was used without any pretreatment. Xylene, acetone, and ethanol were supplied by Kanto Chemical Co., Inc. (Tokyo, Japan). These compounds were used without further purification.

### 2.2. Preparation of PDS Thin-Layer Samples

Before the synthesis of the thin PDS layer samples, the silicon wafers were washed in ethanol and acetone, and then dried under argon gas at ambient temperature (approximately 20 °C) for 30 min. Next, the PDS thin layers were mounted on the silicon substrates (or wafers) by spin coating 2.2 wt%, 6.7 wt%, and 20 wt% of PHPS/xylene solutions (named Samples 1, 2, and 3, respectively) at 6000 rpm using a spin-coater (MS-A150, Mikasa Co. Ltd., Tokyo, Japan). Subsequently, the samples were cured at 60 °C for 1 h and allowed to stand for 7 days at 20 °C. They were then stored in a box under low-humidity and dust-free conditions for seven days at ambient temperature (approximately 20 °C). The PP/PDS/Si substrate samples were prepared by sandwiching the synthesized PDS thin layers between the Si substrates and the PP plate.

### 2.3. Fourier-Transform Infrared (FT-IR) Measurements

Samples for FT-IR spectroscopic analysis were also prepared by removing a section from each sample previously prepared for the NR measurements. The FT-IR spectra were measured with an FT/IR-4100ST (Nihon Bunko Co. Ltd., Tokyo, Japan) system, equipped with an attenuated total reflectance (ATR) unit (PRO670H-S, Nihon Bunko Co. Ltd., Tokyo, Japan). The wavenumber range and resolution were 700–4000 and 4 cm^−1^, respectively. Each spectrum was determined from an average of 64 scans, and all the measurements were performed at ambient temperature.

### 2.4. Neutron Reflectivity Measurements and Data Analysis

The NR measurements were performed using a polarized neutron reflectometer (BL17 SHARAKU, Tokai, Japan) with a horizontal scattering geometry installed at the Materials and Life Science Experimental Facility (MLF) in J-PARC [18]. The incident beam power of the proton accelerator was 500 kW for all the measurements. Pulsed neutron beams were generated in a mercury target at 25 Hz and the NR data were measured using the time-of-flight (TOF) technique. The wavelength (*λ*) range of the incident neutron beam was tuned to approximately 1.1–8.8 Å for the unpolarized neutron mode and 2.4–8.8 Å for the polarized neutron mode by a disk chopper. The covered scattering vector (*Q*_z_) range was 0.008–0.15 Å^–1^, where *Q*_z_ = (4π/λ) sinθ (θ represents the angle of incidence). A 20 mm beam footprint was maintained on the sample surface by using six different types of incident slits. The specular reflection at *Q*_x_ and *Q*_y_ ≈ 0 was obtained by cutting the *Q*_x_ and *Q*_y_ scattering, which are contributed from the rough surface using two neutron slits. The TOF neutron data were collected by a ^3^He gas tube detector without spatial resolution. All the measurements were taken at ambient temperature. The MLF uses the event recording method as a standard data acquisition system [19]. The data reduction, normalization, and subtraction procedures were performed using a program installed in BL17 SHARAKU. Motofit software [20] was used to fit the NR profiles with the least-squares approach to minimize the deviation of the fit. The thickness (*t*, Å), SLD (*ρ*), and Gaussian roughness (*σ*) were evaluated using Motofit. The Fourier-transformed spectra of all the data were also obtained using Motofit.

The PNR measurements were also performed using a BL17 SHARAKU polarized neutron reflectometer. Figure 1 shows the scattering geometry for the NR and PNR studies. The scattering vector *Q*_z_ was parallel to the *z*-axis, which was normal to the film surface. The magnetic moment of the polarized neutron was aligned normal to the scattering plane and parallel to the sample surface. Appendix A shows a schematic outline of the polarized neutron reflectometer SHARAKU. The polarizer and analyzer with Fe/Si multilayer structure provided a wavelength band from 2.0 to 8.8 Å. The magnetic field was applied vertically to the incident neutron beam and spin polarization by a 1-Tesla magnet. The minimum guide field required to prevent the beam depolarization was added to the neutron flight path using a guide coil that was developed at J-PARC MLF. A two-coil spin flipper and a Mezei spin flipper were used as spin flippers (SF) before and after the sample. A neutron beam with a wavelength of 2.4 Å < *λ* < 8.8 Å was provided with a constant polarization efficiency of approximately 98.5%, as measured by the analyzer.

In this study, to decrease the spin-incoherent hydrogen-scattering background in the NR profile, the spin-coherent and incoherent scattering intensities were calculated using a polarization analysis technique [21]. Here, the relationships between the coherent scattering counts *N_C_* and the spin-incoherent scattering counts *N_SI_* can be expressed as follows:(1)$$N_{c}\left(\mathrm{incoherent}\;\mathrm{neutron}\;\mathrm{intensity}\right)=\frac{1}{4}\;\left[\frac{I^{+}+I^{-}}{T^{++}+T^{+-}}+\frac{3(I^{+}-I^{-})}{P_{N}\left(T^{++}-T^{+-}\right)}\right]$$
(2)$$N_{SI}\left(\mathrm{incoherent}\;\mathrm{neutron}\;\mathrm{intensity}\right)=\frac{3}{4}\;\left[\frac{I^{+}+I^{-}}{T^{++}+T^{+-}}+\frac{\left(I^{+}-I^{-}\right)}{P_{N}\left(T^{++}-T^{+-}\right)}\right]$$
where *I*^+^ and *I*^−^ indicate the NR profiles measured with SF off (+,−) and SF on (+,+), respectively. *T*^++^ and *T*^+−^ indicate the transmission ratios of the analyzer for the polarized neutrons. *P_N_* indicates the polarization of the incident neutrons.

## 3. Results

### 3.1. FT-IR Analysis of PDS Thin-Layer Samples

FT-IR measurements and peak assignment of the synthesized PDS were carried out to determine the composition of the synthesized PDS thin layers on the Si substrate. Although some peaks attributed to the absorption of SiO_2_ were observed in the FT-IR data of the mid-sized and thick PDS layer samples, there were no peaks in the FT-IR spectrum of the thinnest PDS layer sample. This could be because of the low amount of SiO_2_ in the thinnest PDS sample. The absorption peaks at 1000 –1100 and 1100–1200 cm^−1^ were mainly owing to the absorption by SiO_2_ [22,23]. The absorption peaks for N–H (3400 cm^−1^) and Si–H (2200 cm^−1^), which could be attributed to the unreacted PHPS molecule [24], were not observed in the spectra (Appendix A). Therefore, these results indicate that the starting PHPS material was mostly consumed during the 7-day curing reaction.

### 3.2. Structural Study of PDS Thin-Layer Samples by NR Analyses

To understand the nanostructure of the single PDS thin layers on the Si substrates, the thickness, density, and roughness of the PDS thin layers with different thicknesses were analyzed using unpolarized NR analysis. Note that the structure of an Si substrate used in Sample-1 was analyzed by unpolarized NR analysis (see Appendix A).

Figure 2a shows the NR profiles of the air–solid reflectivity data for single PDS thin layers with different thicknesses on the Si substrate. The periods of Kiessig fringes of the thinner PDS samples were clearly longer than those of the thicker PDS samples. This indicates that the thickness of the PDS layer increased as a function of the concentration of the PHPS, and the order of the thicknesses was Sample-1 < Sample-2 < Sample-3. Figure 2a also shows the fitting results of these data using the Motofit program. As a naturally oxidized SiO_2_ thin layer was present on the surface of the Si substrates, a two-layer model (PDS/SiO_2_/Si) was employed to fit the obtained NR profiles. However, good fitting curves could not be obtained in the higher *Q*_z_ region when the two-layer model was used to fit the NR profiles. This indicates that an unanticipated additional thin layer existed under or over the PDS layer. Therefore, we employed a three layer model, named Model-1 (PDS (low density)/PDS/SiO_2_/Si) and a three-layer model, named Model-2 (PDS/PDS (low density)/SiO_2_/Si) to fit the obtained NR profiles. As a result, good fitting curves could be obtained even in the higher *Q*_z_ region using Model-1. The symbols in Figure 2a represent the observed NR profiles, while the solid lines represent the calculated NR profiles determined from the structural models. Appendix A summarizes the structural parameters obtained from this analysis. The obtained SLD values of 2.18 × 10^−6^ Å^−2^ were consistent with the published data obtained from the NR methods [5,25]. However, the obtained SLD values of 1.90 × 10^−6^ Å^−2^ for Sample-1 were clearly low compared with those of the others. As this change in the SLD values reflects a change in the density of the PDS layer, it can be suggested that the density of the synthesized PDS thin layer of SiO_2_ was lower than that of the thicker PDS layers, as well as that of the natural SiO_2_ (SLD value was approximately 3.47 × 10^−6^ Å^−2^). The roughness of the air/PDS (low density) surface and PDS (low density)/PDS interface in Sample-1 were larger than those in Sample-2 and Sample-3. Considering the obtained structural parameters, the porosity of the resulting PDS would increase in the thinner (approximately 10 nm) films. As the PHPS precursor had an average molecular weight of approximately 10,000, the radius of the PHPS precursor was estimated to be approximately 2 nm, which was the radius of cytochrome c that had an average molecular weight of approximately 11,000 [26]. This indicates that the curing efficiency of the PHPS would deteriorate because only two or three PHPS precursor layers were stacked on the surface of the substrate, and the PHPS precursors could not approach each other. In addition, all the samples had a low-density PDS area with a thickness of approximately 50 Å on the surface of the sample. Because a sufficient amount of ammonia and xylene gases was generated from the inside of the PHPS polymer during the curing process, the gas volatilization would lead to porosity at the surface of the synthesized PDS layer. Although a low-density PDS layer was formed on the surface of the samples, the densities of the synthesized PDS thin layers were uniform in the depth direction except the low-density PDS area. Therefore, it can be concluded that all the PDS thin layer samples had a low-density PDS layer on their surfaces, and Sample-1, which was a starting material for the preparation of a PP/PDS/Si substrate sample, had a certain thickness of the PDS (approximately 11 nm) and a low density PDS layer (approximately 4 nm) on the surface of the sample.

### 3.3. Structural Study of PP/PDS/Si Substrate Sample by PNR Analyses

To investigate the structure of the PDS thin layers under deep-buried conditions using a nondestructive method, we next, performed PNR measurements of the PP/PDS/Si substrate sample. Because the sample exhibited large incoherent scattering, we carried out polarization analysis of the data, aiming to decrease the spin-incoherent hydrogen-scattering background of the PP plate in the reflectivity profiles.

Figure 3a shows the NR profiles obtained from the *I*^+^ and *I*^−^ data. It is clear that the PP/PDS/Si substrate sample exhibited large incoherent scattering, and the incoherent scattering intensity became even larger compared with the coherent scattering intensities in the *Q*_z_ > 0.14 Å^−1^ region. H atoms of the polypropylene molecules would mainly contribute to the incoherent scattering because the PDS and Si substrate contained only a small amount of hydrogen. Therefore, neutron polarization analysis was carried out to eliminate the spin-incoherent scattering components experimentally to avoid smearing out the characteristic oscillations emanating from the thin layer structure. Figure 3b shows the NR profiles before and after the “spin-incoherent hydrogen-scattering background (BG) subtraction” using Equations (1) and (2). The amplitude of Kiessig fringes in the higher *Q*_z_ region increased after subtracting the contribution of the spin-incoherent hydrogen-scattering background. Thus, it was confirmed that the coherent and spin-incoherent scattering components were clearly separated through the polarization analysis. We performed a fitting analysis for the data in Figure 4a using the Motofit program. There were no significant differences in thickness and roughness obtained from the fitting analysis of the before and after background subtraction data. However, the obtained SLD values of the PDS (low-density) and PDS layer in the “after background subtraction data” (0.73 and 1.93 × 10^−6^ Å^−2^) were lower than those in the “before background subtraction data” (0.83 and 2.10 × 10^−6^ Å^−2^) (see Appendix A). This indicates that the background subtraction affected not only the shape of the NR profiles but also the results of the fitting analysis. As the Kiessig fringe amplitude can be related to the neutron contrast between the substrate and the sample layer, the correct SLD values could be obtained by the background subtraction. Since the slope of the NR profile is related to the roughness of the sample surface and/or interface, the roughness values were slightly changed by the background subtraction. In addition, the low SLD value of the PDS (low-density) layer compared with that of Sample-1 indicates that the bulk PP plate penetrated into the PDS (low-density) layer owing to the sandwiching of the silicon-substrate and PP. Assuming that the PP plate simply penetrated into the PDS layer, from an SLD value of 0.73 × 10^−6^ Å^−2^, the mixing ratio of SiO_2_ to PP molecules in the low-density PDS area was estimated to be 1:1. In addition, although the total layer thickness (the low-density PDS + high-density PDS) of the PP/PDS/Si-substrate sample became slightly thinner than that of Sample-1, the PP plate and Si substrate were separated by the high-density PDS layer even after the sandwiching process. It can be concluded that the fine nanostructure of the PDS thin layer in the deep-buried condition could be elucidated by nondestructive methods of polarized NR analysis, and the results suggest that the synthesized PDS with a uniform layer of approximately 11 nm was sufficient to separate the upper layer and the underlying substrate.

### 3.4. Fourier Transform (FT) Analysis Using Background-Subtracted Reflectivity Profiles

The Fourier transformation process of the reflectivity is one of the useful methods for obtaining model-independent (more objective) information concerning the real-space structure. The thicknesses of the thin layer samples were evaluated by the positions of the peaks in the FT of the X-ray and NR profiles [27,28,29]. Note that the results of FTs simply tell us the layer thicknesses in the sample; however, they generally do not give the layering order. In the PP/PDS/Si substrate sample, the total thickness between the PP plate and Si substrate could be analyzed by the peak position of the FT. For the thickness analysis of the PDS thin layer, the NR data shown in Figure 3b were analyzed by FT with a *Q*_z_ range of 0.06–0.20 Å^−1^ using the Motofit program.

Figure 5 shows the FT of the NR data shown in Figure 3b. There are distinguishable peaks in the FT spectra at 172 Å (before the BG subtraction) and 159 Å (after the BG subtraction), respectively. It is important to note that in this case, the period of Kiessig fringes strongly depends on the total thickness of the sample. According to the problem, the FT cannot discriminate between these layers (low-density PDS, high-density PDS, and SiO_2_). As the total thickness obtained by the fitting analysis was 157 Å, the obtained peak position in the FT spectrum of the “before BG subtraction data” would significantly disagree with the result of the fitting analysis. In addition, the shape of the peak in the “before BG subtraction data” became broader. These results indicate that the NR profiles, which contain strong neutron background scattering, would sometimes lead to an incorrect determination of the structure of thin layer samples even in the FT analysis. Note that similar results were obtained by changing the *Q*_z_ range of the FT (see Appendix A).

Therefore, it can be concluded that the NR polarization analysis for separating the coherent from incoherent scattering for organic samples has a significant advantage not only for fitting analysis of NR profiles, but also for layer-thickness analysis using the FT.

## 4. Conclusions

We analyzed the structure of synthesized PDS thin layers using the NR technique and clarified that all the samples had a low-density PDS area with a thickness of approximately 50 Å on their surfaces. Because the thickness of the low-density PDS area would be of the same size as the PHPS precursor molecule, the surface layer that was derived from the outermost surface of the PHPS precursor would have a low density and a porous structure compared to the bulk PDS layer. To elucidate the function of the PDS thin layers in the deep-buried condition, we performed NR polarization analysis to decrease the spin-incoherent hydrogen-scattering background from the H atoms of the PP molecules. This method elucidated that the total layer thickness (low-density PDS + high-density PDS) became slightly thinner after the sandwiching process; however, a high-density PDS layer still remained between the PP plate and the Si substrate. The results of the FT analysis also indicated that the synthesized PDS layer remained between the PP plate and Si substrate in the deep-buried condition. From the viewpoint of the layer density, the density of the PDS layer was lower than that of natural SiO_2_. The synthesized PDS layer had a sufficiently high mechanical strength to maintain a uniform PDS layer. These results imply that a PHPS precursor size (approximately 4 nm) of the PDS thin layer would not have enough mechanical strength to prevent the penetration of the upper layer material. According to these results, it can be concluded that the NR method with background subtraction by polarization analysis yields very accurate results for the fitting and FT analysis of buried thin layer sample data. The structural analysis using the NR with the polarization analysis can enable the investigation of fine structures of buried thin-layers, and the results can lead to a better understanding of the nano-layers and interface structures of organic polymer compounds.

## Figures and Tables

**Figure 1 polymers-12-02180-f001:**
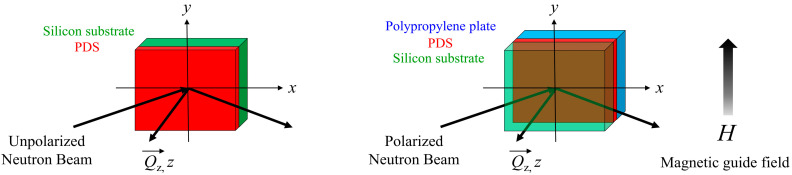
Scattering geometry for neutron reflectivity (NR) and polarized neutron reflectivity (PNR) studies.

**Figure 2 polymers-12-02180-f002:**
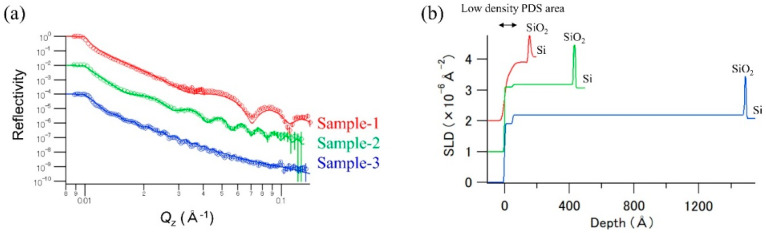
(**a**) NR profiles of the perhydropolysilazane (PHPS)-derived silica (PDS) thin-layer samples (Samples 1, 2, and 3); the circles represent the experimental data, and solid lines represent the best-fit calculated NR profiles; the profiles are vertically shifted to distinguish between them; (**b**) neutron SLD profiles of the PDS layer samples calculated from the obtained structural parameters; the profiles are vertically shifted to distinguish between them.

**Figure 3 polymers-12-02180-f003:**
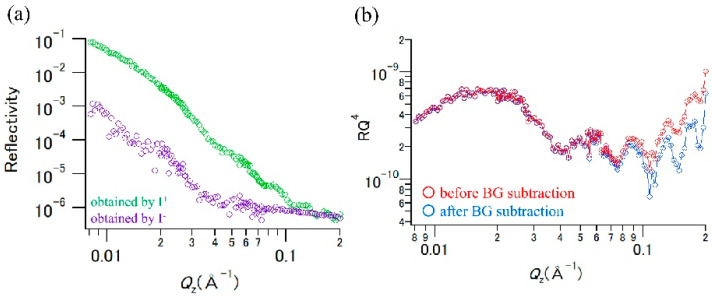
(**a**) Neutron reflectivity profiles of the polypropylene (PP)/PDS/Si substrate sample obtained by *I*^+^ (green) and *I*^−^ (violet) data; (**b**) NR profiles before (red) and after (blue) spin-incoherent hydrogen-scattering background (BG) subtraction; the data are plotted as *RQz*^4^ versus *Q*_z_ to enhance the visibility of the Kiessig fringes.

**Figure 4 polymers-12-02180-f004:**
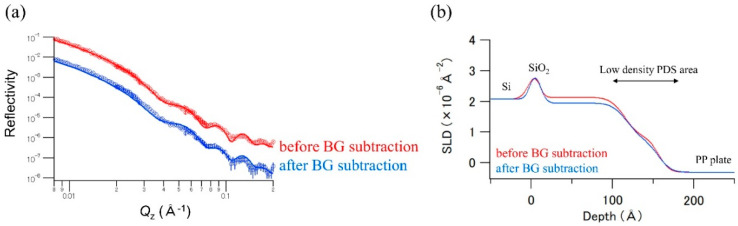
(**a**) NR profiles of before (red) and after (blue) spin-incoherent hydrogen-scattering background (BG) subtracted data; the circles represent the experimental data, and the solid lines represent the best-fit calculated NR profiles; the profiles are vertically shifted to distinguish between them; (**b**) neutron SLD profiles of the PDS layer samples calculated by obtained structural parameters; red line means the neutron SLD profiles obtained from the before BG subtracted data, blue line means the neutron SLD profiles obtained from the after BG subtracted data.

**Figure 5 polymers-12-02180-f005:**
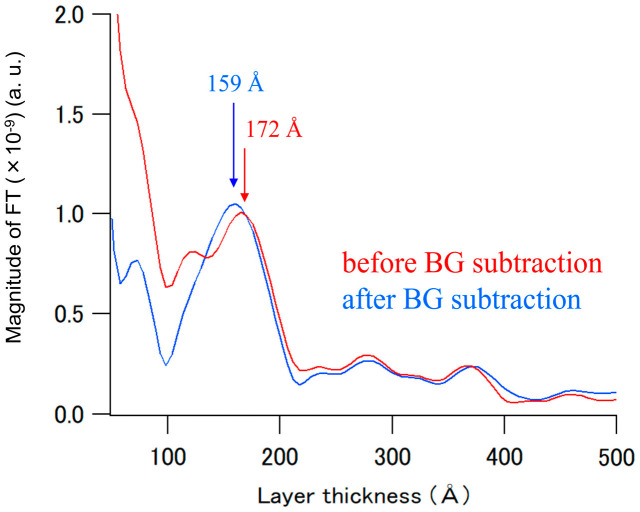
FT of the NR data shown in Figure 3b.

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
