# Peer review of "Fine-Structure Analysis of Perhydropolysilazane-Derived Nano Layers in Deep-Buried Condition Using Polarized Neutron Reflectometry"

_polymers, 2020, doi:10.3390/polym12102180_

Round 1

Reviewer 1 Report

The article (Polymers-933772) focuses on the characterization technology of structure analysis of a nanometric layer in a bulk material. It is a useful topic, since the background scattering originating from the sample matrix may influence the results of the structure analysis. This article introduces a novel method of fine-structure analysis of PDS thin layers using neutron reflectivity technique with polarization analysis, and it is suitable for the journal of POLYMERS. The article is well-prepared and well-organized. I therefore recommend to accept the publication. However, for the benefit of the reader, some minor revisions are suggested to be considered. 1) The fitted parameters of figure 2 and figure 3 were both given. However, in figure 1, the solid lines represent the best-fit calculated NR profiles. It is also better to provide the expression of the fitted if possible, or else, the fitted method should at least be given. 2) What is the difference between the two lines shown in Figure 4(b)? Please also explain the meaning of the two lines. In conclusion, a revision is suggested before its publication.

Author Response

  We are glad to know that Reviewer 1 and Reviewer 2 found our manuscript that was submitted to Polymers “very interesting” and we are grateful for the valuable comments given by all the two reviewers.

  In the current submission (Polymers) we have made changes to the previous manuscript which are indicated by track changes. And below, we have commented on all the questions raised by the two reviewers on the previous version and made changes in the manuscript and the Supplementary data accordingly. 

  Comment 1: The fitted parameters of figure 2 and figure 3 were both given. However, in figure 1, the solid lines represent the best-fit calculated NR profiles. It is also better to provide the expression of the fitted if possible, or else, the fitted method should at least be given.

  Response1: We provided all fitted parameters in the manuscript or Supplementary Material. Please confirm them. Since the fitted method (the expression of the fitted equation), e.g., the Fresnel reflection coefficient and χ square analysis, were shown in the reference 20, we’d like to omit these parts.

  Comment 2: What is the difference between the two lines shown in Figure 4(b)? Please also explain the meaning of the two lines.

  Response2: We wish to thank the Reviewer for this comment. We have added the following sentence to clarify the point “Red line means the neutron SLD profiles obtained from the before BG subtracted data, blue line means the neutron SLD profiles obtained from the after BG subtracted data.”.

Reviewer 2 Report

This paper studied a deep-buried polymer thin-film using polarized neutron reflectometry. The technique can be potentially highly valuable for the polymer/surface science community thanks to its non-destructive nature and ability to probe in-situ. I believe this paper is worthy published if the authors can address the following questions:

  1. Line 47-50. The authors claim “x-ray/neutron beam passes through the sample matrix” can increase “undesired scattering.” While this is generally true, it should not be ignored that the roughness at interfaces makes a big (if not larger) contribution to the background. Eg. A double-side polished wafer has very litter scattering.
  2. Line 86-87. More details are needed for “the PP plate”? What’s the thickness and surface finish? How is it prepared?
  3. Line 129-130: The two equations are identical.
  4. Line 152: it should be stated that the density profiles are vertically shifted, the same in Figure 3.
  5. Line 184-186: the authors claim that “the layers were uniform in the depth direction,” which contradicts the previous claim of “porosity at the surface.” The uniformity of thin film is crucial for reflectivity analysis with the box model the authors were using.
  6. Line 218-219: It’ll be helpful if the authors can give some insight on how the background subtraction affects the fitting.
  7. Line 247-249: the total thickness is 172 and 159 Angstroms, while the NR fitting yield about 147 Angstroms for both samples. It should be pointed out that the FT of NR (or the Patterson function) has limitations when used on non-uniform thin-films. The NR fitting shows the two PDS layers are very different from each other in terms of scattering electron density (in addition to the ~10 Angstroms SiO2), making the FT of NR unreliable.
  8. If possible, please add error bars to the reflectivity curves. And how long did it take to measure each NR curve?
  9. Supplementary: Figure S2. It’s confusing that the authors use red I+ in the bottom figure, while the sub-caption states, “measurement of I-“.

Author Response

  We are glad to know that Reviewer 1 and Reviewer 2 found our manuscript that was submitted to Polymers “very interesting” and we are grateful for the valuable comments given by all the two reviewers.

  In the current submission (Polymers) we have made changes to the previous manuscript which are indicated by track changes. And below, we have commented on all the questions raised by the two reviewers on the previous version and made changes in the manuscript and the Supplementary data accordingly.

  Comment1: Line 47-50. The authors claim “x-ray/neutron beam passes through the sample matrix” can increase “undesired scattering.” While this is generally true, it should not be ignored that the roughness at interfaces makes a big (if not larger) contribution to the background. Eg. A double-side polished wafer has very litter scattering.

  Response1: We wish to thank the Reviewer for this comment. The reviewer pointed to the background scattering from the rough surface. Since the Qx and Qy scattering which are contributed from the rough surface were cut by neutron slits in this experiment, we believe the Qx and Qy scattering did not contribute significantly to the background scattering intensity of the experiments. We have added the following sentence “The specular reflection at Qx and Qy ≈ 0 was obtained by cutting the Qx and Qy scattering which are contributed from the rough surface using two neutron slits.” into the experimental section.

  Comment2: Line 86-87. More details are needed for “the PP plate”? What’s the thickness and surface finish? How is it prepared?

  Response2: The thickness of the PP plate and supplier name were specified in “2.1 Materials” (3 mm thickness and As-One Co., Ltd., (Tokyo, Japan)). The PP surface was not treated, and it has rough surface (μm order routhness). We have added the following sentence “The PP plate was used without any pretreatment.” into the experimental section.

  Comment3: Line 129-130: The two equations are identical.

  Response3: We wish to thank the Reviewer for this comment. We made a big mistake. We the exact equation into eq. (1).

  Comment4: Line 152: it should be stated that the density profiles are vertically shifted, the same in Figure 3.

  Response4: We wish to thank the Reviewer for this comment. The profiles were shifted for a better view. We have added the following sentence “The profiles are vertically shifted to distinguish between them.” into the figure captions.

  Comment5: Line 184-186: the authors claim that “the layers were uniform in the depth direction,” which contradicts the previous claim of “porosity at the surface.” The uniformity of thin film is crucial for reflectivity analysis with the box model the authors were using.

  Response5: In the previous study, we found the porosity structure of the PDS layer at the coating surface and in the bulk by Scanning Electron Microscope (SEM) studies. So, we firstly assumed that the PDS layer is uniform even at the coating surface. However, the NR results elucidated that the synthesized PDS thin layer has a low-density PDS area. The result was unexpected for us. According to the comments, we changed the sentence as “the densities of the synthesized PDS thin layers were uniform in the depth direction except the low-density PDS area”.

  Comment6: Line 218-219: It’ll be helpful if the authors can give some insight on how the background subtraction affects the fitting.

  Response6: The amplitude of Kiessig fringes strongly depends on the neutron contrast between the substrate (or air) and the sample layer. We mention it in Line 220-221. In addition, the slope of the NR profile is related to the roughness of the sample surface/interface. Therefore, we have added the following sentence “Since the slope of the NR profile is related to the roughness of the sample surface and/or interface, the roughness values were slightly changed by the background subtraction.” into the manuscript.

  Comment7: Line 247-249: the total thickness is 172 and 159 Angstroms, while the NR fitting yield about 147 Angstroms for both samples. It should be pointed out that the FT of NR (or the Patterson function) has limitations when used on non-uniform thin-films. The NR fitting shows the two PDS layers are very different from each other in terms of scattering electron density (in addition to the ~10 Angstroms SiO2), making the FT of NR unreliable.

  Response7: We wish to thank the Reviewer for this comment. Considering the scattering length density values of the sample, the period of Kiessig fringes strongly depends on the total thickness of the sample. Therefore, we have added the following sentences “It is important to note that in this case, the period of Kiessig fringes strongly depends on the total thickness of the sample. According to the problem, the FT cannot discriminate between these layers (low-density PDS, high-density PDS, and SiO2).” into the manuscript.

  Comment8: If possible, please add error bars to the reflectivity curves. And how long did it take to measure each NR curve?

  Response8: We wish to thank the Reviewer for this comment. We added error bars to the NR profiles.

  Comment9: Supplementary: Figure S2. It’s confusing that the authors use red I+ in the bottom figure, while the sub-caption states, “measurement of I-“.

  Response9: Neutron spin is flipped at the downstream side Spin Flipper. Therefore, I- data could be obtained using the system. For better understanding of readers, we used blue I- in the bottom figure.

Thank you again for your comments on our manuscript. I trust that the revised manuscript is suitable for publication.